# Analysis of the Effect of Photo and Hydrodegradation on the Surface Morphology and Mechanical Properties of Composites Based on PLA and PHI Modified with Natural Particles

**DOI:** 10.3390/ma15030878

**Published:** 2022-01-24

**Authors:** Karolina E. Mazur, Patrycja Bazan, Aneta Liber-Kneć, Julia Stępień, Alan Puckowski, Adrian Mirowski, Stanisław Kuciel

**Affiliations:** 1Faculty of Materials Engineering and Physics, Institute of Materials Engineering, Tadeusz Kosciuszko Cracow University of Technology, al. Jana Pawła II 37, 31-864 Cracow, Poland; karolina.mazur@pk.edu.pl (K.E.M.); patrycja.bazan@pk.edu.pl (P.B.); 2Faculty of Mechanical Engineering, Institute of Applied Mechanics and Biomechanics, Tadeusz Kosciuszko Cracow University of Technology, al. Jana Pawła II 37, 31-864 Cracow, Poland; aliber@pk.edu.pl (A.L.-K.); stepienjulia98@gmail.com (J.S.); 3Bioliwer Technologies, Górki 3A, 82-500 Kwidzyn, Poland; alan@bioliwer.com (A.P.); adrian27th@gmail.com (A.M.)

**Keywords:** cellulose, walnut shell flour, photodegradation, contact angle, low-cycle tests

## Abstract

Biodegradable polymer materials are increasingly used in the packaging industry due to their good properties and low environmental impact. Therefore, the work was performed on the injection molding of the bio-based composites of polylactide (PLA) and polyhydroxyalcanates (PHI) modified with two phases: reinforcing (walnut shell flour and cellulose) and coloring (beta carotene and anthocyanin). The produced materials were subjected to wide mechanical characteristics—tensile, flexural, and fatigue tests. Additionally, the influence of photo and hydrodegradation on the change of the surface structure and mechanical properties of the composites was assessed. The addition of natural fillers contributed to the improvement of the stiffness of the tested composites. PHI composites withstood a higher number of cycles during cyclic loading, but the stress values obtained in the static tensile test were higher for PLA composites. Moreover, a clear change of color was observed after both the photo and hydrodegradation process for all tested materials; however, after the degradation processes, the filler-modified materials underwent greater discoloration. For the composites based on PHI, the type of degradation did not affect the mechanical properties. On the other hand, for PLA composites, hydrolytic degradation contributed to a higher decrease in properties—the decrease in tensile strength for unmodified PLA after photodegradation was 4%, while after hydrodegradation it was 24%.

## 1. Introduction

Alarming information about the increase in the amount of solid waste, environmental pollution, and the depletion of crude oil has been appearing for several years. Therefore, the European Union (EU) has decided to take the initiative and create legal regulations that will limit the production of disposable products made of plastics. However, the current COVID-19 crisis has seriously tested these plans as there is now a greater need for single-use plastics, while reusable alternatives are being restricted to prevent contamination with the virus [1]. Therefore, both industry and scientific communities are trying to develop biodegradable materials with high strength properties, both for long-term use and for single applications.

High demand for plastic products is observed in every sector of the global industry, e.g., in construction or automotive, but above all in packaging applications, which account for 40% of the total demand for plastics [2]. Other scientists also note the problem of the overproduction of petrochemical plastic packaging [3]. Therefore, it is important to replace petrochemical polymer composites with bio-based ones. In the case of packaging materials, it is important that the materials have appropriate mechanical properties, but also the appropriate transfer of water vapor, oxygen, and/or carbon dioxide, which are factors that determine the speed of many food degradation reactions (oxidation, microbiological development, physiological reactions, etc.) [4].

Two of the most common biodegradable polyesters are polylactide (PLA) and polyhydroxyalkanoate (PHI) polymers. They are bio-based, biodegradable polymers with high strength properties comparable to petrochemical polymers [5]. These materials are biodegradable under certain composting conditions and in other environments, such as seawater or compost heap. These are materials that are very often modified due to their high price and to improve thermal and mechanical properties. It has been mainly assumed that in the modification of biocomposites, natural fillers are used in order to leave the material one hundred percent biodegradable. In addition, the assumptions of the EU, which try to introduce a circular economy, make natural fillers, including waste, resources with added value that can be used to obtain new ingredients and materials [6]. It is estimated that the agricultural sector produces 140 billion tons of biomass, which is waste during food production [7]. The addition of natural waste particles to biocomposites will be a cradle-to-cradle concept and will contribute to promoting the circular economy. One of the most frequently produced natural waste is lignocellulose-containing particles, and its addition to composites based on PLA and PHI has already been researched [8,9]. Mainly lignocellulosic additives increase Young’s modulus and lower tensile strength [10]. Lignocellulosic additives are an excellent alternative to synthetic fibers because some of the lignocellulosic fibers have similar values for specific properties, such as Young’s modulus divided by density. The specific Young’s modulus for E-glass fibers is 26.9–28.0 GPa/gcm^3^, while for sisal, frame and bamboo is 6.5–30.8, 27.3–81, and 50–67.9 GPa/gcm^3^, respectively [7]. In addition, the production of natural fibers requires about 60% less energy than the production of E-glass [11].

In order to increase the attractiveness of products made of polymer composites, they are more and more often modified with dyes or pigments. However, in the case of packaging that comes into contact with food, additional requirements are imposed. Packaging should be safe for the consumer and should not release harmful substances into food [12]. Therefore, in the case of materials intended for contact with food, plasticizers, antioxidants, light, and thermal stabilizers, lubricants, antistatic agents, lubricants, nanoparticles, and some chemical compounds such as bisphenol A, bisphenol A diglycidyl ether, phthalates, and primary aromatic amines should not be used [13]. 

Natural dyes, such as anthocyanin, green tea extract, B-carotene, chlorophyll, curcumin, lutein, etc., are increasingly added as color additives to polymeric materials [14,15]. Therefore, the aim of this work was to produce hybrid polymer materials based on PLA and PHI with the addition of both the reinforcing phase in the form of natural particles (walnut shell flour and cellulose) and the coloring phase (anthocyanin and beta carotene). Basic strength properties such as tensile and flexural strength, as well as fatigue properties have been determined. Additionally, the produced composites were subjected to photo and hydrodegradation. Such comprehensive research will show the impact of natural additives on materials with potential use in the packaging industry, but also contribute to determining the life cycle of such materials. Detailed information about the simultaneous effect of the strengthening and coloring phases for natural fillers is not available in the literature.

## 2. Materials and Methods

### 2.1. Materials

Two thermoplastic biopolyesters supplied by Nature Plast (Ifs, France) were used as matrixes. Their properties are summarized in Table 1. 

Two additives were added to each composite, modifying its mechanical properties and surface color. Natural particles were added as fillers influencing the mechanical properties: cellulose Arbocel P290 (JRS Pharma, Rosenberg, Germany) and walnut shell flour UNG 200 (JRS Pharma, Rosenberg, Germany). As for color additives, the following were added: beta carotene E160a (Bart, Slupno, Poland) (orange) and anthocyanin E163 (Food Colors, Piotrkow Trybunalski, Poland) (blue). The amount of the reinforcement phase was 10 wt% and the colorant 2 wt%. 

Standard dumbbell samples were injection-molded on a KM 40-125 Winner Krauss Maffei (Munich, Germany) injection molding machine. For PLA-based composites, the injection parameters were as follows: mold temperature: 25 °C, zone 1: 140 °C, zone 2: 150 °C, zone 3: 160 °C, nozzle temperature: 170 °C, screw speed: 100–175 rpm and back pressure: 1800 bar. Whereas the injection parameters for PHI were as follows: mold temperature: 25 °C, forming zones: 140 °C (I), 145 °C (II) and 150 °C (III) and 160 °C (IV), screw speed 50 rpm and back pressure 600 bar. 

In order to remove moisture from the test materials, the materials were subjected to a drying treatment prior to injection. The base materials were dried in a DRYMAX primus E30-70-M molecular dryer (Vienna, Austria): PLA—6 h at 80 °C and PHI—40 °C for 4 h, while the natural additives were dried in an oven for 48 h at 80 °C, regardless of their type. The types and acronyms of the produced composites are summarized in Table 2.

### 2.2. Testing Methods

#### 2.2.1. Morphology and Structure Characterizations

Scanning electron microscopy (SEM) was carried out with a (JEOL JSM5510LV, Tokyo, Japan) operating in a low vacuum at 20 kV. In order to increase the conductivity, the samples were covered with a thin layer of gold using an auto vacuum coater (Cressington, Watford, UK).

After photodegradation and hydrodegradation, visual analysis was realized by visual observation and photographing by Keyence (Osaka, Japan).

#### 2.2.2. Contact Angle Measurement

Another important parameter to evaluate the interaction between the material and the environment is the contact angle that is related to the surface wettability, a key parameter, especially for packaging materials. The measurements of a contact angle were performed with a sessile drop method using the optical goniometer (Advex Instrument, Brno, Czech Republic) and the corresponding software SeeSystem. A drop of 0.5 µL of distilled water (Biomus, Lublin, Poland) was dropped perpendicular to the material surface, and the drop profile image was captured. The software automatically calculated the value of the contact angle from the drop profile image based on the height and width analysis of the drop. Each sample surface was subjected to ten measurements. The value of the contact angle was the average of ten values.

#### 2.2.3. Mechanical Tests

The tensile test was carried out on a universal testing machine MTS Criterion Model 43 (Eden Prairie, MN, USA) at ambient temperature (+23 ± 2 °C) and 50% relative humidity (ISO 527). The displacement rate was fixed to 10 mm/min and MTS axial extensometer was used. The flexural strength and modulus were determined on a Shimadzu universal testing machine (Kyoto, Japan) with a crosshead speed of 5 mm/min (ISO 178). The data were averaged on five samples.

#### 2.2.4. Photodegradation

The produced composites were photoaged in a photodegradation instrument ColorCab5 ANTICORR (Gdansk, Poland) using the light color TL84, i.e., 4100 K, which simulates typical lighting inside shops, offices, supermarkets, and exhibitions. The samples were conditioned for 1000 h according to ISO 3668 standard. The samples were immersed in continuously flowing distilled water at a temperature of 40 °C and periodically illuminated with a xenon lamp. Two xenon lamps were used in the research with a power of 1.8 kW mounted above the tested samples, with an exposure area of 1150 cm^2^ and a radiation range of 300–400 nm. After 1000 h of degradation, the specimens were cleaned off the water and dried by paper.

#### 2.2.5. Hydrodegradation

The hydrothermal degradation was carried out in physiological saline solution (2% NaCl) at 38 °C. The ISO 62:2008 standard was adopted as a hydrothermal environment, modifying the temperature and water specification to the desired aging condition. The degradation was carried out for 59 days and the weighing of the samples was carried out at the following intervals: 1, 7, 14, 21, 30, 39, 45, 52, and 59 days. After each extraction, the samples were dried with paper to remove surface water. The water absorption was measured at each stage of the experiment according to the following Formula (1):(1)$$M_{t}=\left[\frac{W_{t}-W_{0\;}}{W_{0\;}}\right]\times 100$$
where *M_t_* stands for the percentage of water content, *W_t_* for the instantaneous weight of the sample, and *W*_0_ for the initial weight of the sample.

After the photo and hydrodegradation, the colorimeter tests were performed (12 measurements per one sample). On the basis of the CIE Lab model, the following distances were determined: *L* (lightness), *a* (color from green to red), and *b* (color from blue to yellow). Based on the specified parameters, the Euclidean distance was calculated using the following Formula (2):(2)$$\Delta E=\sqrt{{\left(\Delta L\right)}^{2}+{\left(\Delta a\right)}^{2}+{\left(\Delta b\right)}^{2}}$$

In order to determine the effect of hydrothermal and photodegradation on mechanical properties, tensile and flexural tests were carried out after 59 days of incubation.

#### 2.2.6. Low-Cyclic Fatigue Tests

Stress-controlled tension–tension fatigue tests were performed at 23 °C using a hydraulic tensile machine Instron 8511.20 (Norwood, MA, USA) at 5 Hz cyclic frequency with a sinusoidal waveform. The testing program consisted of several cyclic load blocks. The force value was selected in proportion to the maximum static force, starting from 30% of the maximum force of each material. For each subsequent load block containing 5000 cycles, the maximum force value was increased by 5%. The test was carried out to the fatigue destruction of the sample or exceeding the limit of displacement. During the tests, mechanical hysteresis loops were recorded and, on their basis, the dissipation energy was calculated. 

## 3. Results and Discussion

### 3.1. Unexposed Samples

#### 3.1.1. Morphological Study

In the case of polymer composites modified with natural fillers, it is very important to assess the morphology of cross-section fractures after the elongation test, including the description of the distribution of fillers in the matrix and their interaction with the matrix. These characteristics are closely related to the mechanical properties of polymer composites. Figure 1 presents images of fractures after the static tensile tests of PLA and PHI composites. As can be observed, the obtained fractures have a brittle nature, as evidenced by the layered arrangement of spherulites. This is a characteristic behavior for brittle materials such as PLA and PHI [16,17]. During the injection molding, an appropriate procedure was used as both the walnut particles and the cellulose were evenly distributed in the matrix and there was no visible agglomeration. The SEM images show that the addition of lignocellulosic fibers without prior treatment resulted in insufficient fiber/matrix adhesion, which is confirmed by the black rings visible around the fibers (marked in red in Figure 1). A more pronounced lack of good fiber matrix interaction is observed in the case of large walnut particles with a size of up to 500 μm. The walnut particles have an irregularly shaped lobe structure and a fibrous structure. On the other hand, cellulose particles are much smaller and better dispersed in the matrix due to the higher specific surface area. Previous research also confirms that the addition of unmodified natural particles results in the lack of adequate adhesion of the fiber matrix and thus a deterioration of the strength properties of materials [18,19].

#### 3.1.2. Contact Angle

In order to assess the effect of the addition of natural fillers on the surfaces of the produced composites, the measurement of the contact angle was carried out, the results of which are presented in Figure 2. With this parameter, the nature of the surface and its reaction to water (hydrophilicity or hydrophobicity) can be assessed. The method is based on the surface tension between liquid-solid interfaces. Materials are usually classified as hydrophilic for contact angles between 0° to 90°, and hydrophobic for contact angles between 90° to 180° [20]. All tested materials show a hydrophilic nature with a contact angle from 63° to 79°. Figure 2 shows that unmodified PLA (79.3° ± 3.5) has a smoother surface than PHI (63.1° ± 3.6), suggesting that there are fewer pores on the surface, making it difficult for the droplets to penetrate into the composite. In the case of PLA-based composites, the addition of lignocellulosic fibers contributed to an increase in the hydrophilicity of the surface, as evidenced by a decrease in the contact angle by over 10°. This phenomenon is commonly known and is related to the introduction of hydrophilic natural fibers into the hydrophobic matrix. In addition, as can be seen in the SEM images (Figure 1), there is insufficient adhesion in composites, which favors the formation of microtunnels, which may also appear on the surface and promote roughness. 

A different situation was observed for the composites based on PHI, the contact angle of which remained at a similar level as for the unmodified PHI. Similar observations were documented by Hassaini et al., who investigated the effect of the PHBV-g-MA compatibilizer on the morphology and properties of poly (3-hydroxybutyrate-Co-3-hydroxyvalerate) (PHBV)/oil husk composites [21]. The addition to PHBV of 10 wt% of the filler slightly reduced the contact angle from 70° ± 0.4 to 67° ± 0.7. Most likely, it is related to the already high hydrophilicity of the material and the addition of only 10 wt% of the filler may not be sufficient for the particles to be present on the surface, and chemical interactions between the hydroxyl groups could occur. 

#### 3.1.3. Tensile and Flexural Tests

Bio-based polymer materials are increasingly used in high-performance materials due to their high properties in relation to density. Bio-based composites not only have high strength properties comparable to petrochemical plastics but are also environmentally friendly. Table 3 shows the results from the static tensile tests (tensile strength and Young’s modulus) and three-point flexural tests (flexural strength and flexural modulus). Despite the biodegradable nature of both matrices, the materials have different properties, which was also documented in our previous work [22]. Composites based on PLA achieved much higher tensile strength values than PHI (over 60%), while composites based on PHI achieved higher values of Young’s modulus, which is in line with the technical description of the used materials provided by the manufacturer. It should be added that the modification of PHI with natural particles contributed to the achievement of similar Young’s modulus values as in the case of PLA-based composites.

The addition of natural particles contributed to the increase in the stiffness of the tested materials, expressed in Young’s modulus. Walnut particles showed higher efficiency as the improvement was 13% and 15%, for PLA/W(c) and PHI/ W(c), respectively, whereas for composites with cellulose, the increase was 10% (PLA) and 1% (PHI). This fact is probably related to the particle size; the key factor for stiffness is not the fiber/matrix interaction, but the particle size. The larger is the size of the fillers, the higher is the stiffness of composites [23]. The addition of natural fillers caused a drop in tensile strength due to the inability to transfer the load from the matrix to the filler. The addition of fillers created additional points that favored the generation of destructive stresses. It should be noted that the second phase aimed at changing the color of the composites could also have an influence on the generation of stresses—more inclusions. The hydrogen bonds between the polar hydroxyl groups of the lignocellulosic particles and the ester groups in the matrices can destroy the dense molecular arrangement in the amorphous region, causing the molecular segments of the polymers to shift when a load is applied. Higher declines in tensile strength value were noted for PHI based composites—20% (W(c)) and 47% (C(a)) than for PLA—17% (W(c)) and 20% (C(a)). 

#### 3.1.4. Dynamic Tests

The determination of the fatigue strength is an important parameter for determining the life cycle of a material. It is primarily an important parameter for two-phase and multi-phase composites, where the decohesion processes taking place (mainly at the phase boundary) appear even during the first load cycles. Conducted studies on the effect of dynamic loads for tested polymers and their composites showed differences in ability to transfer the load depending on the polymer type (Table 4). For PHI composites, a higher number of cycles was observed (55,036 for PHI/C(a)) compared to PLA composites (17,409 for PLA/C(a). However, the values of the transferred cyclic load before fatigue failure were higher for PLA composites, which is related to the higher static tensile strength for these composites. The calculated dissipation energy as the surface area of the last recorded hysteresis loop before failure (Figure 3) was higher for composites filled with cellulose and anthocyanin compared to walnut shell flour and carotene. This may be due to better adhesion observed in SEM images (Figure 1) for lignocellulosic fibers and matrix.

The type of fillers used affected the load-bearing capacity, and in the case of composite with walnut shell flour and carotene, the number of cycles to destroy the material turned out to be the lowest, even lower than for polymers.

### 3.2. Exposed Samples

#### 3.2.1. Surface Color

In the case of polymeric materials whose field of application is to be the packaging industry, an important parameter is to determine the influence of various environmental conditions the behavior of the materials surface. Figure 4 and Figure 5 present microscopic images showing the surfaces of PLA (Figure 4) and PHI (Figure 5) composites before the degradation process and after photo and hydrodegradation. Unmodified PLA is transparent, and it is on the basis of this material that much clearer colors were obtained than in the case of milky PHI. The addition of beta carotene resulted in the production of yellow-orange materials (PLA/W(c)—amber yellow and PHB/W(c)—honey yellow). On the other hand, anthocyanin has tinted the materials a pinkish-purple color.

The unmodified PLA lost its transparency after hydrodegradation and changed its color to milky, while after 1000 h of photodegradation it was still transparent. In the case of PHI composites, the opposite situation was met: photodegradation was more important. The addition of natural particles caused the samples to become discolored after the degradation processes. Additionally, both degradation processes led to the formation of white efflorescence. More white dots were seen for the composites with added cellulose for both PLA and PHA. Halász and Csóka attribute the white spots to the bacteria that appeared during the biodegradation of PLA composites with cellulose [24]. In their research, the degradation was carried out in industrial composting conditions at 58 °C and 60% humidity for 45 days. The second cause of the white efflorescence is the use of undemineralized water in the hydrodegradation process, which may have contributed to the formation of lime and chlorine deposits on the surface of the samples. However, this issue should be verified with additional research. 

When evaluating the surface of polymeric materials, the Euclidean distance (ΔE) is often used. ΔE describes the quantitative relationship of the colors shown in Figure 6 [25], and its values are computed by formula (2). In the range 0 < ΔE < 1, there is no difference in color change. In the range 0 < ΔE < 1, only experienced observers will notice the difference. In the range 2 < ΔE < 3.5, the difference can be noticed by the average observer. In the range 3.5 < ΔE < 5, there is a distinct color difference, while in the range 5< ΔE, the colors are perceived as completely different [26].

Based on the results presented in Table 5, it was found that all samples had a color change after both the photodegradation process and the water absorption test. The lowest color change was observed for unmodified materials—PLA ΔE = 0.85, PHI ΔE = 7.33 after photodegradation and PLA ΔE = 3.64, PHI ΔE = 2.06 after photodegradation. The addition of cellulose to PLA-based composites has a higher effect on the color change of the surface of materials than the addition of walnut shell flour, while in PHI composites the situation is reversed, for both photo and hydro degradation tests.

Moreover, PLA-based composites underwent more color change during hydrolytic degradation, while a greater color change for PHI composites was noted during photodegradation. Similar results were obtained by Latos-Brazio and Masek, where in their work they studied the influence of natural dyes on the behavior of PLA and polyhydroxybutyrate (PHB) composites [27]. The following natural dyes were used: beta carotene, chlorophyll, curcumin, and lutein. The materials have undergone aging processes (thermo-oxidation, UV, and weathering). The polyesters with the natural dyes beta carotene, curcumin, and lutein were characterized by a clear change of color under the influence of UV radiation, increased temperature, and weather conditions. Less pronounced color changes were observed for the polyesters with chlorophyll, but the color changes in this material were visible to the observer. 

#### 3.2.2. Water Absorption

In the case of polymer composites reinforced with particles containing lignocellulose, it is very important to determine the water absorption and its influence on the mechanical properties. The composite water absorption mechanisms include diffusion through the matrix, capillary through the natural fibers, or movement through porosities in the matrix or at the fiber-matrix interface. Consequently, the water absorption depends not only on the relative hydrophilicity of the fiber and the matrix but also on the fiber-matrix interface and the morphology of the composites [28]. Figure 7 shows the results of the sample mass change during hydrolytic degradation in a physiological saline solution at 38 °C for up to 59 days. Our previous research shows that PLA-based composites have a higher water absorption capacity than PHI-based composites [22]. Composites based on PHI are subject to a higher degree of degradation, and thus the material does not gain weight but disintegrates. It was also confirmed in these tests by the surface quality assessment presented in Figure 5. For unmodified composites, there was the lowest weight gain, not exceeding 1%. Because PLA and PHI are hygroscopic in nature, they can absorb approximately 1% water [29]. The addition of natural fibers increased the absorption capacity of the composites. Fibers containing lignocellulose are hydrophilic, due to the high number of hydroxyl groups that are contained in cellulose and hemicellulose, so their addition to polymer composites favors the absorption of water. Water molecules form hydrogen bonds with free –OH groups. Additionally, as shown in Figure 1, microchannels in the composites are visible, which contributed to an even higher water absorption capacity of modified composites. In the case of both matrices, the addition of cellulose contributed to the highest water absorption: 5% for PLA and 2% for PHI, after 59 days of incubation.

#### 3.2.3. Mechanical Test after Photo and Hydrodegradation

Figure 8 presents the results of mechanical tests (tensile strength and Young’s modulus) after photo and hydrodegradation. For the composites based on the PHI matrix, the type of degradation did not affect the mechanical properties; the decrease in properties was at a similar level. On the other hand, for PLA composites, hydrolytic degradation contributed to a higher decrease in properties; the decrease in tensile strength for unmodified PLA after photodegradation was 4%, while after hydrodegradation it was 24%. The degradation rate of the modified PLA and PHI samples was higher than the degradation rate of unmodified composites. This was due to the higher water absorption capacity. Lignocellulose fibers, which show high hydrophilic properties, change their dimensions (swell) under the influence of water, which leads to the generation of additional stresses creating microcracks. Along with the propagation of microcracks, the capillarity and transport of water droplets through the microchannels are activated, which causes the particles to detach from the matrix and do not have the ability to transfer loads from the matrix to the fibers. The highest drops in properties for both PLA and PHI composites were recorded for composites with walnut particles. For PLA/W(c), the decrease was 10% and 42% (tensile strength) and 34% and 23% (Young’s modulus) after photodegradation and hydrodegradation, respectively. On the other hand, for PHI/W(c), the differences in relation to the unmodified polymer were −25% and 25% (tensile strength) and −30% and 30% (Young’s modulus) after photodegradation and hydrodegradation, respectively. It was related to the particle size of the walnut, the adhesion of which was insufficient (Figure 1), and additionally, water absorption contributed to the increase in the lack of fiber/matrix interaction.

## 4. Conclusions

The work successfully produced hybrid composites based on biodegradable PLA and PHI materials. The composites were strengthened with natural particles, walnut shell flour, and cellulose particles and stained with beta carotene and anthocyanin. For all tested composites, an increase in the value of Young’s modulus was observed and the highest was for composites on PHI with the addition of walnut particles and beta carotene. Due to the lack of sufficient fiber/matrix interaction, the tensile strength results deteriorated, which is a common problem in composites where there is both a hydrophobic and a hydrophilic phase. 

The addition of natural fibers contributed to the increase of water absorption and the highest absorption was observed for composites based on PLA with the addition of cellulose. Both photo and hydrodegradation contributed to a change in the color of the sample surface and a decrease in mechanical properties. The smallest color change was observed for the non-filled materials, where this change could be monitored only by an experienced observer. On the other hand, the addition of natural fillers caused the samples to become discolored after the aging processes. PLA composites were more sensitive to color change during hydrolytic degradation, while a greater color change for PHI composites was noted during photodegradation. For PHI composites, the type of degradation did not affect the mechanical properties; the decrease in properties was at a similar level. On the other hand, for PLA composites, hydrolytic degradation contributed to a higher decrease in properties; the decrease in tensile strength for unmodified PLA after photodegradation was 4%, while after hydrodegradation it was 24%. 

## Figures and Tables

**Figure 1 materials-15-00878-f001:**
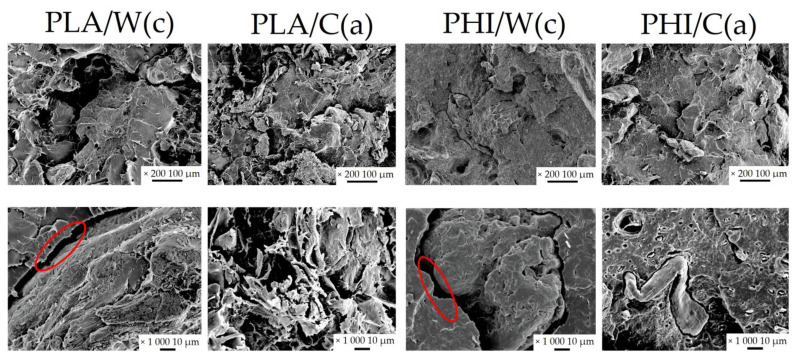
SEM images of PLA and PHI composites.

**Figure 2 materials-15-00878-f002:**
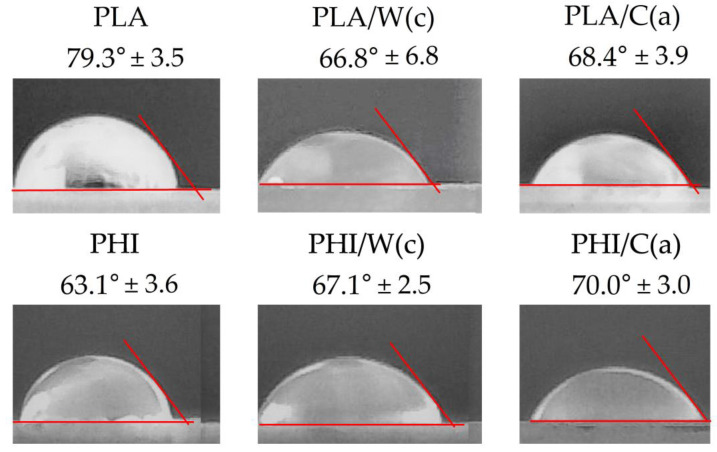
Contact angle and shape of water drops on the PLA and PHI and its composites.

**Figure 3 materials-15-00878-f003:**
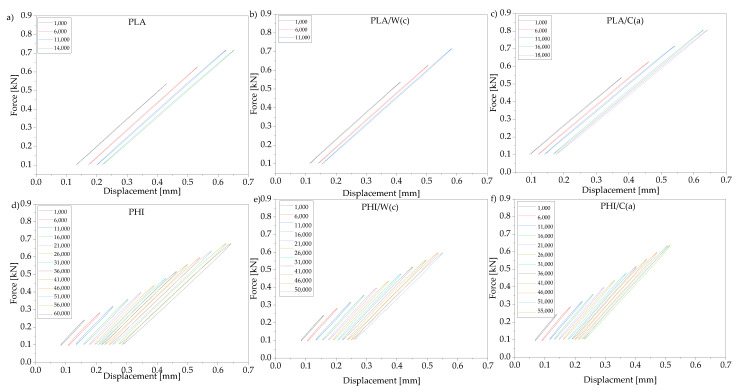
Hysteresis loop obtained during low-cycle fatigue tests for PLA and PHI composites: (**a**) polylactide, (**b**) polylactide+walnut shell flour+carotene, (**c**) polylactide+cellulose+anthocyanin, (**d**) polyhydroxyalkanoates, (**e**) polyhydroxyalkanoates+walnut shell flour+carotene, (**f**) polyhydroxyalkanoates+cellulose+anthocyanin.

**Figure 4 materials-15-00878-f004:**
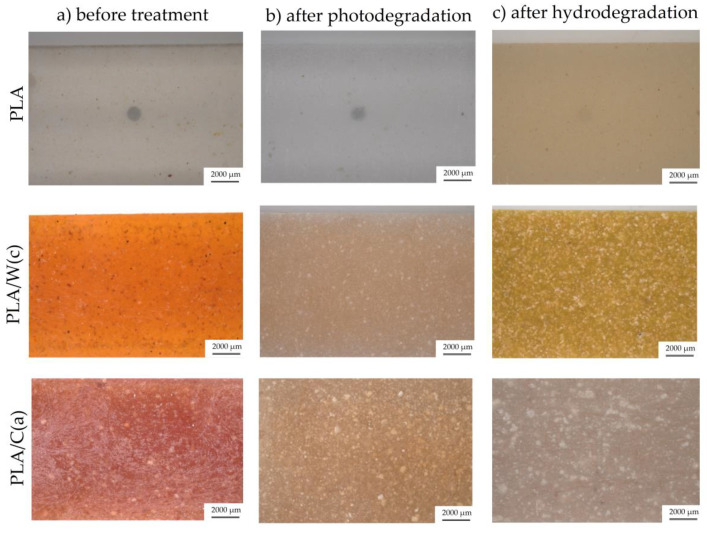
Optical images of PLA composites: (**a**) before treatment, (**b**) after photodegradation, and (**c**) after hydrodegradation.

**Figure 5 materials-15-00878-f005:**
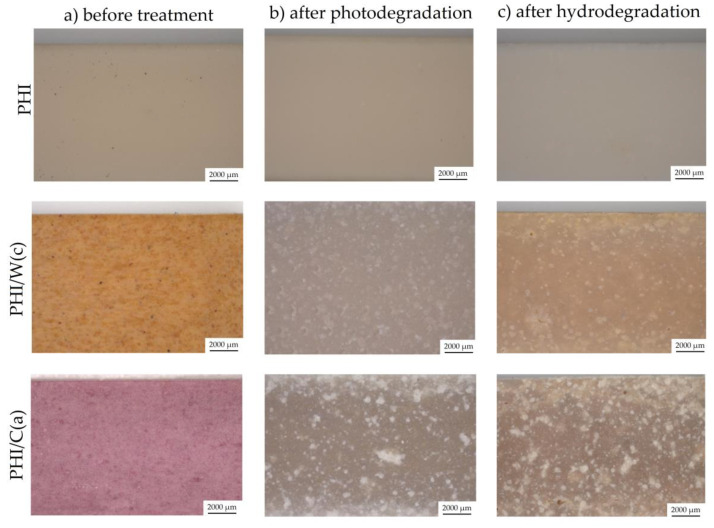
Optical images of PHI composites: (**a**) before treatment, (**b**) after photodegradation, and (**c**) after hydrodegradation.

**Figure 6 materials-15-00878-f006:**
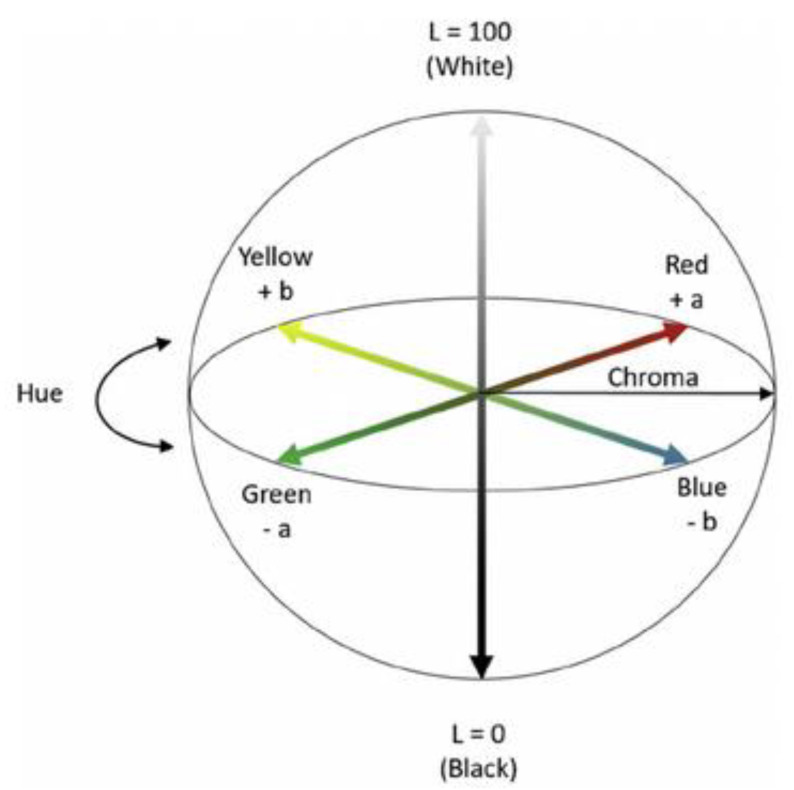
The CIE Lab color model.

**Figure 7 materials-15-00878-f007:**
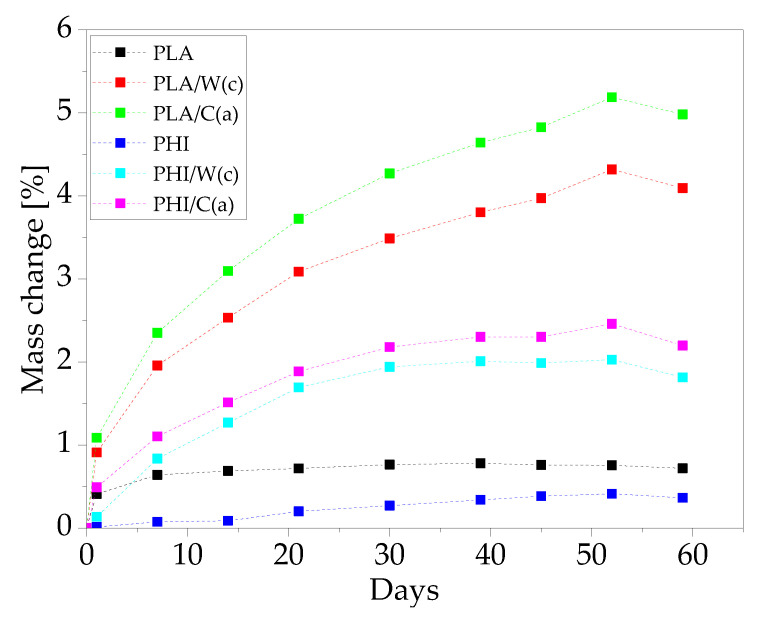
Mass gain of PLA and PHI composites.

**Figure 8 materials-15-00878-f008:**
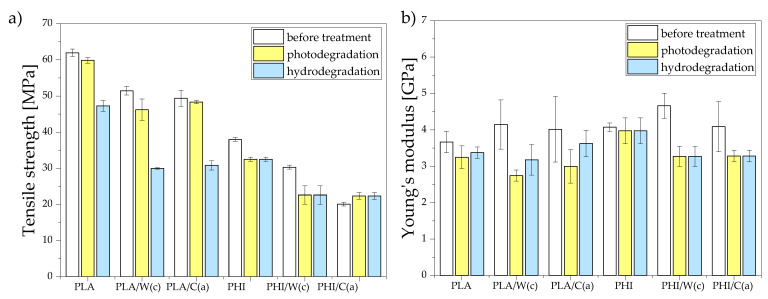
Mechanical properties after photodegradation and hydrodegradation: (**a**) tensile strength and (**b**) Young’s modulus.

**Table 1 materials-15-00878-t001:** Properties of the used matrix.

Properties	Polylactide (PLI005)	Polyhydroxyalcanoates (PHI 003)
Density (g/cm^3^)	1.25	1.24
Fluidity index (g/10 min) 190 °C/2.16 kg	25–35	15–30
Tensile modulus (MPa)	3,500	4,200
Tensile elongation at break (%)	4	4
Charpy impact test, without notch (kJ/m^2^)	22	5
Thermal resistance (°C) (HDT B)	53	134

**Table 2 materials-15-00878-t002:** Acronyms of the produced composites and their compositions.

Index	Compositions
PLA	100% Polylactide
PLA/W(c)	Polylactide + 10 wt% walnut shell flour + 2 wt% carotene
PLA/C(a)	Polylactide + 10 wt% cellulose + 2 wt% anthocyanin
PHI	100% Polyhydroxyalcanoates
PHI/W(c)	Polyhydroxyalcanoates + 10 wt% walnut shell flour + 2 wt% carotene
PHI/C(a)	Polyhydroxyalcanoates + 10 wt% cellulose + 2 wt% anthocyanin

**Table 3 materials-15-00878-t003:** Results from tensile and flexural tests.

Samples	Tensile Strength [MPa]	Young’s Modulus [GPa]	Flexural Strength [MPa]	Flexural Modulus [GPa]
PLA	62.0 ± 1.1	3.7 ± 0.3	102.8 ± 3.1	3.7 ± 0.1
PLA/W(c)	51.5 ± 1.2	4.2 ± 0.7	87.7 ± 3.9	3.9 ± 0.1
PLA/C(a)	49.4 ± 2.2	4.0 ± 0.9	83.7 ± 4.0	4.0 ± 0.2
PHI	38.0 ± 0.5	4.1 ± 0.1	68.9 ± 2.9	3.5 ± 0.2
PHI/W(c)	30.3 ± 0.6	4.7 ± 0.4	56.5 ± 1.5	3.8 ± 0.1
PHI/C(a)	20.1 ± 0.5	4.1 ± 0.7	57.0 ± 2.0	4.5 ± 0.3

**Table 4 materials-15-00878-t004:** Dynamic properties: dissipation energy, maximum force, and number of cycles.

Samples	Dissipation Energy (mJ)	Maximum Force (kN)	Number of Cycles
PLA	2.4	0.72	13,617
PLA/W(c)	2.1	0.72	11,907
PLA/C(a)	4.0	0.81	17,409
PHI	7.5	0.68 *	60,000
PHI/W(c)	5.1	0.64	50,191
PHI/C(a)	5.6	0.68 *	55,036

* without fracture.

**Table 5 materials-15-00878-t005:** Euclidean distance of PLA and PHI composites after photo and hydro degradation.

Samples	After Photodegradation	After Hydrodegradation
PLA	0.85	3.64
PLA/W(c)	7.86	10.63
PLA/C(a)	10.92	30.50
PHI	7.33	2.06
PHI/W(c)	36.60	24.02
PHI/C(a)	15.02	13.29

## Data Availability

The raw/processed data required to reproduce these findings cannot be shared at this time due to technical limitations. Specific (or example) data may be sent on request by the corresponding author.

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
