# Peer review of "Analysis of the Effect of Photo and Hydrodegradation on the Surface Morphology and Mechanical Properties of Composites Based on PLA and PHI Modified with Natural Particles"

_materials, 2022, doi:10.3390/ma15030878_

Round 1
Reviewer 1 Report
This work reported the influence of natural particles on the properties of PLA or PHI composites, which is of practical significance. However, some issues should be addressed before it can be accepted for publication. Some specific comments are listed as follows. Is there a standard for hydrodegradation analysis? The intervals are not consistent. The description in water contact angle part is not accurate. It should be water contact angel that indicates the hydrophilicity or hydrophobicity. Please add the explanation on the phenomena that cellulose improved the Young’s modulus of the PLA while it showed negligible influence on the PHI. Statistic analysis should be added. The authors claimed the importance of replacing the present plastic packaging materials with biodegradable alternatives, thus, the reported works in this area should be appropriately acknowledged, such as Carbohydrate Polymers. 2021, 271, 118425.Author Response
The responses were attached in a separate file.
Reviewer 2 Report
There is one comment, correct the spelling of 3.1.2 "Contact Angel" as "Contact angle".
Author Response
The responses were attached in a separate file.

Reviewer 3 Report
I really appreciated reading your work, thank you. Please see my suggestions, regarding a few minor mistakes and some English writing, in the attached pdf file.

Author Response

(The authors gave the same response as above.)

Round 2
Reviewer 1 Report
My previous concerns have been properly addressed.